

# 600 years of wine must quality and April to August temperatures in Western Europe 1420–2019

Christian Pfister[1], Stefan Brönnimann[2], Andres Altwegg[3], Rudolf Brázdil[4], Laurent Litzenburger[5],
Daniele Lorusso[6], Thomas Pliemon[7]

[1]Oeschger Centre for Climate Change Research, University of Bern, Switzerland christian.pfister@unibe.ch
[2]Oeschger Centre for Climate Change Research, Institute of Geography, University of Bern stefan.broennimann@unibe.ch.
[3]Dr. Ing. Agr. ETH, altwegg@greenmail.ch
[4]Institute of Geography, Masaryk University, Brno, Czech Republic, brazdil@geogr.muni.cz
[5]Université de Lorraine, Nancy, CRULH, l.litzenburger@free.fr
[6]Università degli Studi di Milano, Italy danielelorusso.unimi@gmail.com
[7]Institute of Physics, Department of Astrophysics and Geophysics, University of Graz, Austria, thomas.pliemon@uni-graz.at

*Correspondence to*: Christian Pfister (christian.pfister@unibe.ch)

**Abstract.** This study investigates the validity of wine must quality as a summer (JJA) temperature proxy between 1420 and 2019 based on expert ratings and quality measurements from Germany, Luxembourg, Eastern France and the Swiss Plateau. The evidence was reviewed according to the best practice of historical climatology. Expert ratings tended to agree with Oechsle density measurements that gradually replaced them from the 1840s. A statistical model calibrated to predict wine must quality from climate data explains 75% of the variance, underlining the potential value of wine must quality as a climate proxy. Premium crops were collected in years of early harvest involving high insolation during maturation, while poor crops resulted from very late harvests in cold and wet summers. An analysis of daily weather types for high and low-quality years after 1763 shows marked differences. On a decadal timescale, average quality was highest from 1470 to 1479, from 1536 to 1545 and from 1945 to 1954. Poor crops were collected in periods prevailing cold and wet summers such as 1453 to 1466, 1485 to 1494, 1585 to 1614, 1685 to 1703, 1812 to 1821 and 1876 to 1936. In the period of enhanced warming after 1990, high qualities became the rule.

## 1 Introduction

Viticulture in the mid-latitudes of Europe is known to be closely linked to weather and climate. Due to the sensitivity of grape yields to the weather in the growing season, the revenues of wine producers (i.e., price multiplied by quantity) and grape quality also vary considerably (Ashenfelter and Storchman, 2016). With only minor other influences, temperature controls the rate of physiological development of the vine (*Vitis vinifera L.*) through budbreak to flowering, setting, veraison





and berry ripening (Gladstones, 2011). Grape harvest dates (GHD) from archives of societies are thus essential for the high-

resolution reconstruction of past climate during the warm season. Pejml (1974) using 19th century wine quality data for Moravia (Czech Republic) concluded that for years with good or excellent wine, the probability of above-mean temperatures in the vegetation period achieved 73%, while in the years with low quality the probability of below-mean temperatures was 77%. Brázdil et al. (2008) analysed wine quality from three 19th and early 20th century series from Moravia (Czech Republic), suggesting that the relations between grape quality and harvest together with climatic conditions could build an

additional proxy to reconstruct the temperature conditions in past times. Andrea Kiss et al. (2011) included grape quality data from the late 18th century in her May–August temperature reconstructions for western Hungary. In fact, whereas the longest homogeneous GHD series goes back to the mid-14th century (Labbé et al., 2019), proxies using the quantity and the quality of wine must are available back to the 11th century for extreme years. In 1043, for example, chroniclers from north-eastern and southern Germany and what is now Belgium complained of "cold, almost wintry rains with strong winds", which

led to a poor harvest of sour grapes and a wet grain crop, resulting in famine (Alexandre, 1987). In recent decades, the relationship between drought and temperature in Western Europe has weakened, because enhanced warming from anthropogenic greenhouse gases can generate the high temperatures needed for early harvests without drought (Cook and Wolkovich, 2016).

This article investigates grapevine-must quality with weather and climate, connecting historical with oenological and

climatological research over the past 600 years. The term "grapevine-must" originates from the Latin "vinum mustum" meaning "young wine" https://www.winenium.com/must-grape-juice (retrieved 5th Ja-23). In the historical past, grapes were crushed manually by foot or by a hand operated wooden press while the freshly pressed juice flowed into a receptacle. "Quality" denotes the amount of sugar in the wine must. To become proper wine, the wine must needs to be stored in barrels, of which the material (oak or metal) contributes to the final quality of the bottled wine.

55       The German physician and natural scientist Gustav Schübler (1787– 1834) was the pioneer of historical oenology in Europe, as this field of research may be called. As a professor of botany and natural history at the German University of Tübingen, he investigated the influence of weather on cultivated plants (Loose, 2022). He compiled information on GHD along with the quantity and quality of wine must harvests in historical documents back to the late Middle Ages to explore "whether the climate of Germany has changed for centuries, whether it has remained the same, or whether these conditions

are perhaps due to greater periods that have so far escaped observation" (Schübler, 1831:3). He thus already considered the three oenological parameters that are used today for climate reconstructions.

       After the warm summers in the late 1940s arousing fear of desertification, climate change became an issue again. Looking for evidence in the past millennium, Karl Müller (1953), who then was director of the State Viticulture Institute in Freiburg Br, compiled the climatologically relevant oeno-climatic sources known for southern Germany standing on

Schübler's shoulders. For the same reasons, the Luxembourg professor Eugène Lahr (1950) may have drawn on ancient data on the quality and quantity of Moselle wines in the Grand Duchy of Luxembourg. Alessandro Rima (1963) investigated the periodicities in a long time-series of grapevine quality and quantity from the Johannisberg Castle situated in Geisenheim



(Hesse). Lauer and Frankenberg (1986) analysed the climatic parameters affecting grape quality and harvest amount in the German federal state of Palatinate (west of the Rhine) over the period 1948– 1980 using multiple regression and a principal component analysis (in German). Their results are summarized as follows: The months from May to August of the harvest year explain the highest variance among all significant influencing factors. In May, soil temperatures matter, in June sunshine becomes important, in July also temperature, while in August drought is the main factor. Conditions in autumn, on the other hand, fall off. In general, cold, low-radiation and damp weather in the phase from May to July of the harvest year increases the acidity of the wine at the expense of the sugar content. Molitor et al. (2016) investigated the impact of seasonal heat on grapevine must quality and quantity in the vine growing region in Luxembourg for the reference period 1854–1885. While wine must quality was highly significant, vine must quantity as an additional predictor variable did not improve model output.

The US oenologist Gregory V. Jones and the environmental scientist Robert E. Davis investigated records of harvest date, harvest quantity and quality using vineyard observations kept from 1938 to 1995 in Bordeaux (France). The two authors concluded that a large part of the annual fluctuations in the three above-mentioned oeno-climatic proxies was controlled by a few large-scale weather events. Cyclonic weather conditions with strong winds and cold fronts reduce grape quantity and quality also delaying the ripening of the berries. On the other hand, warm high-pressure conditions allow early large harvests with high sugar content to mature. In other words, late harvests were always small and sour, while early harvests were often large and of high quality (Jones and Davis, 2000).

The Italian historian Daniele Lorusso (2013, 2018) investigated the extent to which GHD can be climatically complemented and differentiated through information on wine must quality. In his innovative PhD, he compiled 26 relevant vintage expert rating time-series for the period of the 18th and 19th centuries. His results tend to agree with those of Jones and Davis (2000), according to which early harvests generally correspond to good and very good wines, and late harvests to sour and poor crops. For "average" cases in the middle time intervals of seasonal differentiations, he obtained information on rainfall and short-term shocks on grape quality besides temperatures. Lorusso also demonstrated significant correlations between grape quality and wine prices for the 19th and 20th centuries. Outside Europe, wine studies in the humanities and social sciences are just a part of food history, alcohol history or both (McIntyre, 2017).

The present study extends the above-mentioned investigations of wine must quality in a western European setting back into the early 15th century. The study raises the question of whether the reconstruction of summer temperatures based on GHD can be improved by including evidence of wine-must quality. The paper is structured as follows. Section 2 presents the sources and discusses their reliability and validity. In section 3 we analyse the statistical properties of regional series before they are merged into an overall series. Section 4 on methods focusses on the climatic interpretation of the overall series. Section 5 reviews the main results and discusses their limitations. The paper closes with a view to further research.





## 2 Sources

### 2.1 Overview of sources

As early as the 11th century, chronicles refer to the sugar content of the wine-must and the size of the harvest in exceptional years by reference to a proxy that had a close connection to their experience (Pfister and Wanner, 2021). Wine was in the past at the same time an everyday drink, a religious symbol, and a cash crop, and its sugar content already mattered for its price (Lorusso, 2018). Reports on wine-must quality became more continuous from the 13th century onwards. Starting in the early 19th century, they were systematically collected for documenting past climate fluctuations.

| Table 1 Survey of sources | | | | | |
|---|---|---|---|---|---|
| | | | | | |
| **Location** | **Alt m** | **Type** | **Period** | **%** | **References** |
| | | | | | |
| 1. Baden-Württemberg De | 200 | E | 1420–1950 | 3 | Schübler, 1831, Pfaff, 1865, Müller, 1953 |
| 2. Remich Lu | 160 | E | 1627–1989 | 3 | Lahr, 1950, S5, S17, 1950–1989 |
| 3. Briedel De | 118 | E | 1644–1989 | 32 | Source S6 |
| 4. Johannisbg De | 95 | E | 1700–1989 | 0 | Rima, 1968, Staab et al., 2001 |
| 5. Swiss Plateau | 380–450 | E | 1529–1880 | 1 | DOI https://doi.org/10.48620/317 provided |
| 6. Remich Lu | 160 | °Oe/Bx | 1839–2019 | 0 | Statec, 1989, Source S17,1988–2019 |
| 7. Ct Schaffhausen Ch | 400–430 | °Oe | 1970–2019 | 0 | BLW, 1970–2019 |
| 8. Neustadt De | 136 | E | 1420–1867 | 3 | Source S4 |
| 9. Eastern France | 200–300 | E | 1720–1879 | 1 | Lorusso, 2013 |
| 10. Canton Zürich (1) Ch | 420–450 | °Oe/Bx | 1938–2019 | 0 | Source S5 1961–69, BLW 1970–2019 |
| **Series 1 to 10** | | | 1420–2019 | | |

Legend; Alt Altitude above sea level. E = Expert opinions, °Oe = degrees Oechsle, Bx: Brix, % = % gaps
1) 1938–1960 District Meilen

Table 1 presents the temporal coverage of the used sources. Following Schübler's (1831) approach, the educationalist Karl Pfaff (1795–1866) published a "Württemberg Wine Chronicle" including quality and quantity data since the 13th century (Source S3). Two years later, the German naturalist and oenologist Friedrich August Dochnahl published a [wine] chronicle for the town of Neustadt an der Weinstrasse (Rhineland-Palatinate, Germany) covering the long period from 1420 to 1867 (Source S4). The most reliable data for the period 1420 to 1537 were drawn from the study for Metz by Laurent Litzenburger (2015). Eugène Lahr (1950) compiled a wine chronicle from 1627 onwards for the Luxembourgish





vine growing region located at the northern border of viticulture in Europe at that time (Fig. 1). His grape quality chronicle
was continued up to 1989 (Source S13). After the hot-dry summers of the late 1940s, Karl Müller (1953) supplemented the
data collected by Pfaff up to 1950 (Source S2). For verification, a wine chronicle of the wine village of Briedel (Rhineland-
Palatinate) from 1644 to present was consulted (Source 6). The longest must quality series based on expert ratings known so
far was drawn from the winery of the Johannisberg Castle from 1700 to 2000 (Staab et al., 2001). Most Swiss sources
originate from the cantons Zürich (Source S7), Schaffhausen (Source S8), Thurgau (Source S9) and St. Gallen (Source S10)
situated in the North-eastern part of the Swiss Plateau. The estimated annual qualities for Germany, France and Luxembourg
and for the Swiss Plateau are contained in the repositorium https://doi.org/10.48620/317 (provided).

From the mid-19th century, expert ratings were gradually replaced by measurements of sugar content. This device
measures the density of the unfermented wine-must before it is stored in barrels. It corresponds to the proportion of dissolved
substances (mainly sugar). Weighing the wine-must yields a standardized quality criterion comparable across regions and
continents. According to current knowledge, the mechanic and goldsmith Ferdinand Oechsle (1774 –1852) from Pforzheim
(Baden, Germany) invented the must scale in 1836. However, he was not the forerunner. In fact, must weighing was
practised in southern Germany since 1754. Schübler (1831) quotes a relevant treatise by the Stuttgart city physician J.J.
Reuss (1773). The device was regularly used from 1839 in the Duchy of Luxembourg for purposes of quality control (Statec,
1989). In the Swiss cantons, the method was introduced at different times starting in the late 19[th] century. In Canton
Schaffhausen, for example, it is documented from 1853. Starting in 1874, the communities had to report the results of the
annual must weighing to the authorities (Source S8), whereas this regulation was introduced in Canton Zürich only after
1938 (Altwegg et al., 2023). From 1970 onwards, the Swiss Federal Office for Agriculture published the cantonal averages
of must weighing according to cultivar (Source S9). Starting in 1995, results are given in degrees Brix (°Bx) (Source S14)

Based on geographical location (Fig. 1), three groups of sources can be identified. Metz (France), Remich
(Luxembourg) as well as Briedel, Neustadt an der Weinstrasse and Johannisberg (Germany) are located at the historical
northern wine-growing border, where the vines are particularly sensitive to warm and cold summers. The Rhineland,
Württemberg and the Swiss Plateau are located farther south, whereas the French communities are situated more to the west.

## 2.2 Source reliability

The best practice of historical source evaluation was applied to assess the reliability of notations on vintage expert ratings. In
addition to transcriptional and printing errors, wine chronicles are likely to have included flawed information from
uncontemporary sources. However, such flaws are difficult to detect. At least, flawed quality data may be tracked down
indirectly, be it by contradiction of added weather reports in the chronicles with those from contemporary sources, or by
cross-checking with evidence from neighbouring regions. Some gaps and inconsistencies were filled with Schübler's (1831)
original data as well as with evidence from the comprehensive study for Metz by Litzenburger (2015). Quasi continuous
contemporary data are available for Swiss cantons from the official annual assessment of wine must prices. In the town of
Zürich, from 1421 onwards, the authorities used to take a sample of wine must from a vineyard near the lake on the eve of





Martinmas (Nov 11th) to estimate the price of wine per bucket, often also specifying the quality (Nabholz, 1908). The vineyards owned by the authority were cultivated by sharecroppers, who had to give half of the yield as rent, while they sold their half on the market and to creditors. The authorities tried to prevent speculation, stabilizing their income in record years

by setting a takeover price while accommodating the sharecroppers in case of crop failures (Source S18). Corresponding official price and quality indications are documented for the cantons of Schaffhausen, Thurgau, St. Gallen and Zurich from different points in time (Sources SA8, SA9, SA10, SA 47 DOI https://doi.org/10.48620/317) provided).

### 2.3 Source validity

Wine [must] quality can be defined as the average rating assigned by wine experts to a vintage (Almaraz et al., 2015). At

present, the cross-correlation between different sets of expert ratings is usually large (Jones et al., 2005; Baciocco et al., 2014). It is assumed that ratings were made by consensus among guilds (?) of local vine-growers who had the necessary knowledge and experience.

To assess the validity of documentary data as climate proxies, the evidence from different independent sources is compared. Correlation of regional vintage rating time-series allows certain conclusions to be drawn in this respect, but the

uncertainties involved cannot be quantified. As chroniclers used quite similar terms to refer to wine-must quality, the source information could be standardized. Following Bassermann-Jordan (1907), it became commonplace to distinguish five quality classes of wine must from 1 (premium) to 5 (nearly undrinkable), often adding a summary description of relevant growing-season weather spells (Lauer Frankenberg, 1986; Lorusso, 2018; Molitor, 2016).

| Table 2 | Size of regional quality classes (expert ratings) 1420–1989 | | | | | | | | | | | |
|---|---|---|---|---|---|---|---|---|---|---|---|---|
| | | | | | | | | | | | | |
| Class | N | % | N | % | N | % | N | % | N | % | Total | % |
| 1 | 45 | 16 | 51 | 16 | 56 | 11 | 20 | 12 | 23 | 7 | **195** | 12 |
| 2 | 64 | 23 | 83 | 26 | 141 | 29 | 30 | 20 | 121 | 33 | **439** | 27 |
| 3 | 94 | 28 | 71 | 22 | 140 | 27 | 39 | 24 | 101 | 31 | **445** | 28 |
| 4 | 48 | 22 | 73 | 23 | 139 | 27 | 37 | 23 | 75 | 23 | **372** | 23 |
| 5 | 38 | 11 | 38 | 12 | 38 | 6 | 33 | 21 | 18 | 6 | **165** | 10 |
| **Total** | **289** | **100** | **316** | **100** | **514** | **100** | **159** | **100** | **334** | **100** | **1616** | **100** |
| | | | | | | | | | | | | |
| | Joh.berg De | | Remich Lu | | Germany | | East France | | Swiss Plateau | | | |
| | 1700–1989 | | 1627–1964 | | 1420–1950 | | 1720–1896 | | 1525–1880 | | | |






Table 2 Size of regional quality classes (expert ratings) 1420–1989.

The allows to compare the relative size of the quality classes among regions according to expert ratings. Overall, the proportion of premium and poor wines is the lowest. On the other hand, the proportion of good wines tends to be overestimated.


The differences involved in table 2 point to the fact that expert ratings may be classified as acceptable approximations.

| Number and region | 1 | 2 | 3 | 4 | 5 | 6 | 7 | 8 | 9 |
|---|---|---|---|---|---|---|---|---|---|
| 1. Baden-Württ. De 1420–1950 | | | | | | | | | |
| 2. Remich Lu 1627–1964 | 0.70 | | | | | | | | |
| 3. Briedel De 1644–1989 | 0.63 | 0.74 | | | | | | | |
| 4. Johannisberg De 1700–1989 | 0.66 | 0.75 | 0.66 | | | | | | |
| 5. Swiss Plateau 1525–1880 | 0.58 | 0.51 | 0.45 | 0.49 | | | | | |
| **6. Remich Lu 1839–1989** | **-0.68** | **-0.90** | **-0.71** | **-0.64** | **-0.54** | | | | |
| **7. Schaffhausen Ch 1938–1989** | N.A. | **0.55** | **0.40** | **0.48** | N.A. | **-0.53** | | | |
| 8. Neustadt De 1420–1867 | 0.84 | 0.67 | 0.59 | 0.61 | 0.54 | N.A. | N.A. | | |
| 9. Eastern France 1720–1896 | 0.61 | 0.53 | 0.50 | 0.51 | 0.52 | -0.56 | N.A | 0.64 | |
| **10. Zürich(1) Ch 1938–1989** | N.A. | N.A. | **0.43** | **0.48** | N.A. | **-0.54** | **0.68** | N.A. | N.A. |

**Table 3.** Correlations between regional series of expert ratings, 1420–1989
Probability rho all correlations p<0.01
Legend: Bold Oechsle density measurements
(1) Meilen 1938–1980

Table 3 presents a statistical comparison of the classified regional series. Values show the coefficient of correlation
among regional series that are all significant. The correspondence among the German, Luxembourg and Swiss expert ratings is highly significant, while the values for the French municipalities located farther West are somewhat lower. It is concluded that the good spatial agreement of the evidence speaks for its validity as a climatic proxy. Note that, measured °Oechsle values are highly correlated with expert ratings from the same location, as the example of the Upper Mosel valley (r -0.92) and that of French data (Lorusso, 2013) shows. Starting from the mid-19th century, expert ratings were gradually abandoned
in favour of Oechsle density measurements. In Switzerland, this was done by canton at different times, whereas in Luxembourg expert ratings were maintained in parallel to measurements until 1989.



# 3 Methods

To check the quality of climate reconstruction over time, the regional expert ratings were compared with grape harvest dates (GHD). This proxy is available in the form of long time-series (Daux et al., 2012). However, it should be noted that the

setting of the vintage date is based on a decision-making process. Due to severe frost or long rains, vintners may start harvesting before the grapes have reached full maturity or in the case of plague or military threats the harvest may be skipped altogether (Garnier et al., 2011). Taking the example of the highly cited "Dijon series" (Chuine et al., 2004), a critical source review may even be needed for contemporary data. The two historians, Thomas Labbé and Fabien Gaveau (2011), discovered that the extremely early harvest date set by the prefect of the department Côte d'Or in the hot summer of

2003 was in fact a bureaucratic artefact. According to the local press, the grape harvest in the Burgundian wine metropolis of Beaune actually began 19 days after the official date.

To compensate for local effects, two long GHD series were used, namely that of Beaune (France) from 1354 to 2018 based on the red Pinot Noir cultivar (Labbé et al., 2019) and that of the Swiss Plateau from 1461 to 2011 (Wetter and Pfister, 2013) based on traditional white grape cultivars, mostly Elbling and Räuschling (Aeberhard, 2005). The vines in

Beaune are grown at altitudes between 220 m and 300 m (a.s.l.), whereas those in the Swiss Plateau are grown at altitudes between 370 m and 500 m. Both series were homogenized dividing the anomalies from the mean by the standard deviation and averaging the result. The resulting composite series named "GHD+" starts in 1461. In addition to GHD, the highly cited 1,250 years long series of maximum latewood density (MXD) data from the Lötschental (Canton Valais) (Büntgen et al., 2006) was integrated as an independent proxy.


| Table 4 | Correlation of GHD+ with must quality and tree-rings | | | | | |
|---|---|---|---|---|---|---|
| | 1420–1530 | 1531–1626 | 1627–1750 | 1751–1880 | 1881–1989 | Mean |
| | | | | | | |
| **Must quality-GHD+** | rho | rho | rho | rho | rho | rho |
| Germany-Luxembg | 0.54 | 0.42 | 0.56 | 0.75 | 0.50 | 0.56 |
| Swiss Plateau | | 0.67 | 0.56 | 0.61 | 0.36 | 0.52 |
| Eastern France | | | | 0.74 | | |
| **Mean** | **0.54** | **0.59** | **0.64** | **0.83** | **0.46** | **0.61** |
| | | | | | | |
| **Tree Rings-GHD+** | rho | rho | rho | rho | rho | rho |
| MXD Lötschental | -0.37 | -0.26 | -0.41 | -0.37 | -0.38 | -0.36 |

Table 4 Correlation of GHD+ series with wine must quality and MXD tree ring data over time
Probability: rho all correlations p<0.01
Legend: GHD+ mean standard deviation of the GHD series of Beaune (France)
(from 1420) and of the Swiss Plateau series (from 1460)



Data: Tree-ring MXD data Büntgen et al., 2006

Table 4 shows the degree of agreement between the GHD+ series with regional wine must quality and MXD tree ring data over time. Coefficients in table 4 are lowest from 1420 to 1524, due to a small number of cases per year and the fact that until 1460 only the GHD series from Beaune is available. Coefficients rise until reaching a maximum in the period 1750 to

1880. Between 1881 and 1989 they are lower, mainly due to a high frequency of wetness related harvest failures (Fig. 6) aggravated by Downy Mildew. This fungal disease, not to be confused with Phylloxera, was brought in from North America causing severe crop loss in wet summers (Altwegg, 2023).

Figure 2 shows scatterplots of GHD+ and wine must quality 1420–1989 (right) and 1751–1880 (left). Correlations between

GHD+ and wine must quality over the whole period of investigation show a relatively large scattering, which is mainly due to low case numbers and the availability of a single GHD series before 1460. On the other hand, the correlation for the period between 1781 and 1880 (Fig. 2) reaches a maximum of 69% whereby the outliers are due to extreme weather events (Table 5).

## 4 The climatic model

230        A model for investigating the influence of growing-season weather on wine must quality needs to build on the vegetation cycle of the vines. The blossoms, after flowering in early summer, develop into berries after fruit set in midsummer. Up to this stage, the development of the vine is essentially controlled by the temperature between April and July. At the beginning of the ripening period, called veraison, in August the berries are still small and hard. Subsequently they soften, change colour and their sugar content increases at the expense of acidity. The ripening process takes about three

to four weeks (Combe et al., 2015). Dry and sunny conditions matter during this period (Lauer and Frankenberg, 1986). The earlier the ripening begins, the greater the chance to get a good crop, as the insolation decreases relatively quickly from mid-August. On the other hand, the berries hardly ripen in a late year.

        Based on this information, we used a statistical model for wine must quality based on monthly local temperature and precipitation. We started by considering temperature and precipitation of all months from April to August as predictor

variables and then proceeded with a backward selection. Variables that were not statistically significant ($p<0.05$) were excluded from the model.

        The meteorological variables for the regression model were taken from the re-analysis (Valler et al., 2023). This is a global monthly climate reconstruction of the period 1421–2008 based on assimilating natural proxy data, documentary data, and instrumental observations into an ensemble of global atmospheric model simulations (Bhend et al. 2023). The generation

of the product closely follows the predecessor product EKF400v2 (Valler et al., 2022), but uses a greatly expanded set of documentary data (Burgdorf et al., 2023) and instrumental observations (Lundstad et al., 2023) and a new ensemble of climate model simulations (Hand et al., 2023). From this data set, which consists of 20 ensemble members, we used the



ensemble mean of the grid point 5.6° E and 47.6° N (see Fig. 1) to represent the area of interest. We calibrated the model in
the period 1781–1880, because instrumental data coverage over Europe for this period is excellent, so that the reconstruction
is of high quality. The remaining years were used for independent evaluation.

Further, we used daily weather type reconstructions for Switzerland back to 1763 from Schwander et al. (2017). The
weather types are based on the official MeteoSwiss weather types CAP9 (Weusthoff et al., 2011), which encompass nine
weather types. For the reconstruction, these were reduced to seven types as there were two pairs of two weather types that
were not easily distinguishable. The weather types were successfully used for addressing, e.g., the decadal variability of
floods (Brönnimann et al., 2019). A brief description of the types is given in Table 6.

| # | Abbreviation. | Description |
|---|---|---|
| 1 | NE | Northeast, indifferent |
| 2 | WSW | West-Southwest, cyclonic, flat pressure |
| 3 | W | Westerly flow over Northern Europe |
| 4 | E | East, indifferent |
| 5 | HP | High Pressure over Europe |
| 6 | N | North, cyclonic |
| 7 | WC | Westerly flow over Southern Europe, cyclonic |

**Table 6.** Description of CAP7 weather types (Schwander et al., 2017). CAP7 Cluster Analysis of Principal Components with
7 types

260   Daily weather type reconstructions offer an additional way to explore the relation between wine must quality and climate. In
this study, we analysed weather type frequency since 1763 for years with particularly high and low quality of wine must.

Figure 3 shows the average sea-level pressure patterns: Types WSW cyclonic, N cyclonic and WSW over S Europe are
cyclonic weather types that are often associated with the passage of a weather system. Types NE indifferent, E indifferent
265   and HP over Europe (which is rather rare) are anticyclonic weather types expected to be related to comparably warm and dry
conditions. (According to Schwander et al., 2017)
CAP7 Cluster Analysis of Principal Components with 7 types

## 5 Results

The presentation of the results begins with an overview of the climatic analysis and then discusses the final wine must
270   quality series and its changes over time. The relation of wine must quality with climate was investigated with the regression
model. The backward selection resulted in a model that retained temperature of all considered months (April to August) as
well as precipitation in April and in August (but not the months in between). The results are summarized in Table 7.



| Series | $R^2_{calib}$ | $R^2_{verif}$ | $T_{Apr}$ | $T_{May}$ | $T_{Jun}$ | $T_{Jul}$ | $T_{Aug}$ | $RR_{Apr}$ | $RR_{Aug}$ |
|---|---|---|---|---|---|---|---|---|---|
| Wine Must Quality | 0.75 | 0.43 | *** | *** | *** | *** | *** | ** | ** |
| Grape Harvest Dates | 0.87 | 0.73 | *** | *** | *** | *** | * | ** | ** |

**Table 7.** Results from the statistical modelling approach for series of wine must quality and grape harvest date. Columns indicate the explained variance in the calibration period (1751–1880), in the verification period (1525 to 1750) as well as the significance of the coefficients (*** denotes p<0.001, ** denotes p<0.01.

The regression model provides an excellent fit to the data, with an explained variance of 75%. The performance in the verification period (1525 to 1750) is lower. This could be due to fewer series contributing to the average in earlier years, or a lower quality of the reconstruction (see also Table 4). Nevertheless, the agreement is still good.

In Fig. 4, we show the observed and reconstructed wine must quality series back to 1420. The reconstruction fits well with the observations also outside the calibration window. However, the model overestimates extremes, which is arguably due to the linearity of the model. It predicts values outside the restricted quality scale. A simple truncation to the interval [1,5] already increases correlations. Variable transformations as in Labbé et al. (2019) might further increase the accuracy.

For comparison, we also applied the same regression model to grape harvest dates and found an even better performance (87% explained variance in the calibration, 73% in the verification period). It should be noted, however, that grape harvest dates entered the reconstructions and thus the two data sets are not entirely independent, although the amount of instrumental data in the calibration period is massive, such that the influence of grape harvest dates on the reconstruction is arguably small.

In Fig. 5, we analysed the daily weather types for five years with excellent wine must quality and five years with very low quality in the period 1763 to 1947. The cyclonic weather types 2 (WSW cyclonic), 6 (N cyclonic) and 7 (W flow over Southern Europe, cyclonic) were more frequent in years with low wine must quality than in years with high quality. Conversely, types 1 (NE indifferent), 3 (westerly flow over Northern Europe) and 4 (E indifferent) dominated during years with high must quality. This follows the expected pattern, except that one might also have suspected more abundant high-pressure days over central Europe (type 5) in the good quality years. We find almost the same number (19 versus 20). This is perhaps due to the fact that this type is rare.

Figure 6 presents the average annual wine must quality in Western Europe 1420 to 2019. Prior to 1988, years with high and low quality fluctuated around the long-term average (2.94). 42% of the top-quality wines were obtained between 1471 and 1616, 36% between 1834 and 1983, while only 5 vintages achieved top marks in the 195 years between 1616 and 1811. In three periods (1469–1513, 1656–1674 and 1940–1978), high qualities were slightly more frequent, whereas poor qualities predominated in the three periods (1559–1589, 1800–1853 and 1888–1929). On a decadal time-scale, quality was highest in



the three decades from 1470 to 1479, from 1536 to 1545 and from 1945 to 1954. Each decade culminated in a remarkable hot extreme, namely 1473, 1540 and 1947. 1473 was probably the hottest year in the Millennium (Camenisch et al., 2020), whereas 1947 was the hottest year between 1719 and 2002 (Pfister and Wanner, 2021). Wines were often undrinkable between 1491 and 1542 and during the 237 year long period between 1713 and 1939. Between 1782 and 1821 this applies to
one in five harvests.

Premium grape musts were obtained in the 30 years 1442, 1471, 1473, 1479, 1483, 1484, 1516, 1536, 1540, 1590, 1599, 1615, 1616, 1669, 1676, 1706, 1753, 1811, 1834, 1846, 1865, 1895, 1911, 1921, 1929, 1947, 1949, 1959, 1971 and 1983, whereas 1947 was probably on top. A bottle of Château d'Yquem was on sale for 2950 euros in 2021 (Pfister and Wanner, 2021). In 1540, grapes were ripe in early August albeit severely suffering from water stress. In many vineyards, the
picking of grapes was interrupted and resumed after a rain-spell in September (Wetter and Pfister, 2013). The late vintage yielded a great amount of honeysweet dirt-cheap wine which led to excessive drinking. Apparently, hundreds of people drank themselves to death (Dochnahl, 1867). In the Prince Bishopric residence in Würzburg, the 1540 premium wine was reserved for guests and members of the Court. The last bottle still survives in the city's civic hospital (Glaser, 2013).

Terrible wines were obtained in the 30 years 1428, 1465, 1491, 1517, 1527, 1529, 1538, 1542, 1627, 1675, 1713,
1716, 1725, 1740, 1763, 1782, 1785, 1792, 1799, 1805, 1809, 1813, 1816, 1817, 1821, 1843, 1860, 1879, 1913 and 1939. Some of them got nicknames. The wine in 1529 was called "Türkenwein" (Turkish wine) because the Turks besieged Vienna in that year, and the sour drink in 1860 became known as "Garibaldi" after the Italian freedom fighter of the time.

The striking characteristic of the entire series (Fig. 6) is the almost continuous increase in must weights with the transition from slow to rapid anthropogenic warming in 1988. Besides the effect of climate change, the increase in quality in
the Swiss Plateau is also due to quantity restrictions imposed by the authorities after several bumper harvests and the delaying of the harvest date due to good weather conditions (Altwegg et al., 2023). Most wine musts obtained in the past 20 years must be classified top compared with past centuries.

Figure 7 compares the evolution of must quality, summer days and hot days 1970–2019. A meaningful comparison of
meteorological parameters with wine must quality since 1970 requires focusing upon a single cultivar. Riesling (Müller-Thurgau) was selected because it was grown in Remich for a long time. The average must weight of this cultivar increased by 24% from the period 1970–1989 to 2000–2019. The number of summer days increased by 75% and the number of hot days tripled over this time-period. Due to enhanced warming, new cultivars were grown at the expense of traditional varieties. The downside of this development includes the spread of heat-loving pests, the higher frequency of rot due to more
warm and humid weather in late summer and drought stress and water shortage in hot summers (Source S17, Weinjahr 2007–2018).



## 5 Conclusions

The study demonstrated that evidence on wine must quality based on expert ratings is a valid summer temperature proxy so long as the documentary evidence is critically evaluated. Considering the effort spent on this task, the question of the added

climatic value of wine must evidence for historical climatology arises. Where GHD are lacking or questionable, wine-must quality may be the most reliable highly-resolved summer temperature proxy, invaluable for establishing drought summers and cold-wet seasons. Prior to the late 15th century, it supports and complements the long Beaune GHD series starting in 1354 (Labbé et al., 2019). Further research should focus on the size of grape harvests which is also related to weather and climate. Economic historians might be interested in wine prices which depended on both the quantity and quality of grape

harvests.

**Author contributions**

Christian Pfister was responsible for conceptualization, data curation, formal analysis, investigation, methodology, resources, validation and writing, original draft preparation, as well as review and editing. Stefan Brönnimann was responsible for meteorological and statistical analysis, modelling, validation, visualization, writing and review. Andres Altwegg was

responsible for data acquisition, data analysis, writing and review. Rudolf Brázdil was responsible for resources and writing. Laurent Litzenburger was responsible for data acquisition and writing. Daniele Lorusso was responsible for data acquisition and writing. Thomas Pliemon was responsible for draft preparation, visualization, and editing.

**Funding**

This research was not funded.

**Data Availability Statement**

Christian Pfister, Stefan Brönnimann, Andres Altwegg, Rudolf Brázdil, Laurent Litzenburger), Daniele Lorusso, Thomas Pliemon. Annual wine must quality data from Germany, Luxembourg, France and the Swiss Plateau for 1420 to 2019. BORIS https://doi.org/10.48620/317 (provided), 2023.

**Acknowledgements**

Stefan Doktor, managing director, Schloss Johannisberg (Germany)
Christian Sieber, Staatsarchiv des Kantons Zürich





**Conflict of Interest**

The authors declare no conflict of interest.

**Abbreviations**

The following abbreviations are used in this manuscript:

BLW: (Swiss) Bundesamt für Landwirtschaft.
°Brix: a measure commonly used to measure dissolved sugar content of an aqueous solution (Wikipedia). Conversion:
https://pdfslide.net/documents/tabelle-di-comparazione-dei-valori-di-comparazione-sim-jaulmes-tabelle.html?page=1 (4 Feb
2023).
CAP7: Cluster Analysis of Principal Components with 7 types.
CAP9: Cluster Analysis of Principal Components with 9 types.
De: Germany.
Ch: Switzerland.
Fr: France
°Oechsle: A measure commonly used to measure dissolved sugar content of an aqueous solution.
GHD: Grape harvest date.
GHD+: mean standard deviation of the GHD series of Beaune (France) (from 1420) and of the Swiss Plateau series (from
1460).
Lu: Luxembourg.

MXD: Maximum Tree Ring Density

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





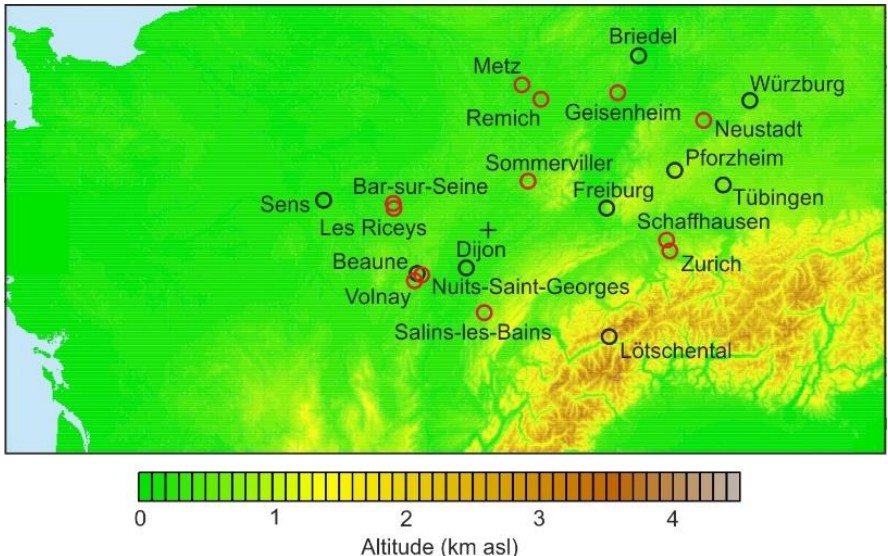

Figure 1: Spatial layout of sources and locations

Red circles: location of sources; black circles: other locations mentioned

in the text. The cross + marks the location of the grid point used for the climatic

model (Sect 3) (based on ETOPO1, NOAA National Centers for Environmental

Information (https://www.ncei.noaa.gov/products/etopo-global-relief-model)).

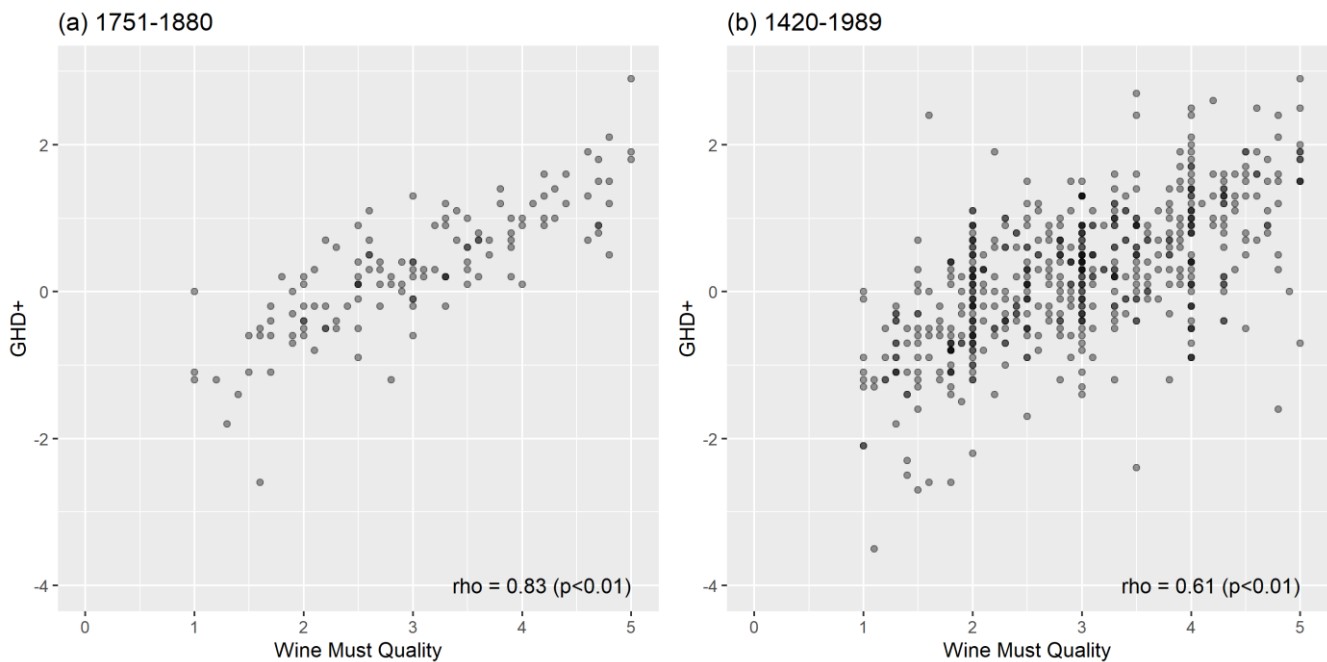


Figure 2. Scatterplots of GHD+ with average wine must quality 1751 to 1880 and 1420 to 1989.

GHD+: mean standard deviation of the GHD series of Beaune (France) (from 1420) and of the Swiss Plateau series (from





1460). Correlations between GHD+ and wine must quality over the whole period of investigation (right, 1420–1989) show a relatively large scattering, which is mainly due to low case numbers and the availability of a single GHD series prior to
1460. On the other hand, the correlation for the period between 1751 and 1880 (left), reaches a high coefficient of determination of 69% whereby the outliers are due to effects of extreme weather (Table 5).

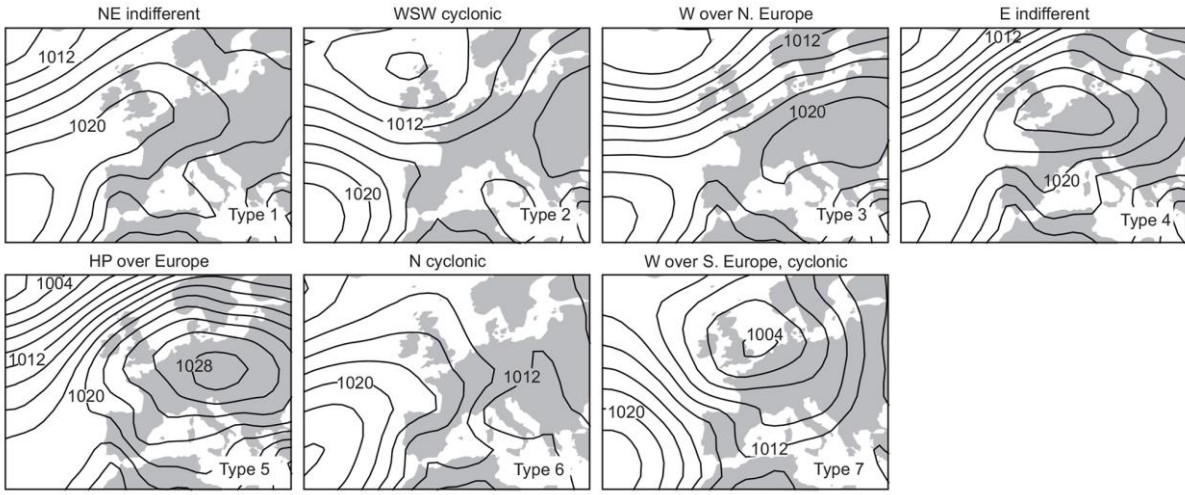

Figure 3, Sea-level pressure averaged for each of the 7 weather types in CAP7 over the warm season
(May–October) for the period 1958–1998 based on 20CRv2c (from Brönnimann et al., 2019)
CAP 7 Cluster Analysis of Principal Components with 7 types.

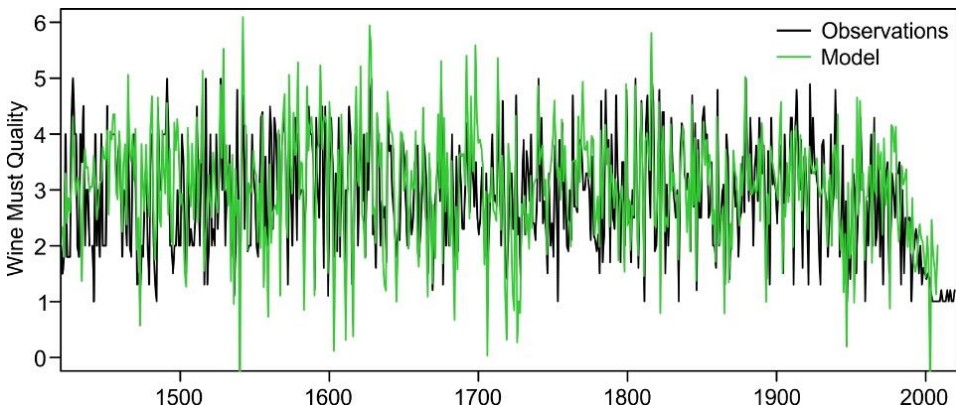

Figure 4. Observed series of wine quality (average) (black) from 1420 to 2019 and series obtained with
a statistical model calibrated in 1781–1800 (green). The model is explained in Sect 3.



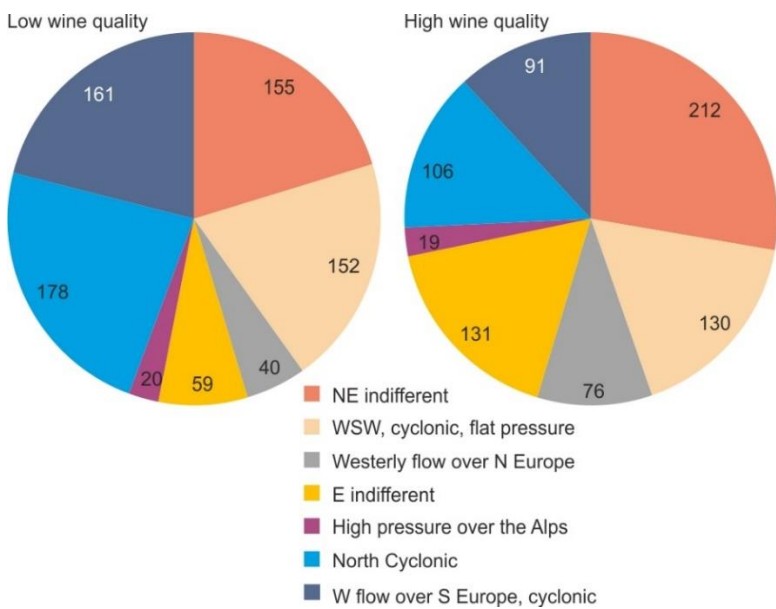

Figure 5. Frequency of Swiss weather types from April to August for years with
low and high wine must quality
Years with low wine must quality (left, 1785, 1799, 1816, 1860 and 1879)
Years with high wine must quality (right, 1811, 1834, 1846, 1947, and 1949)

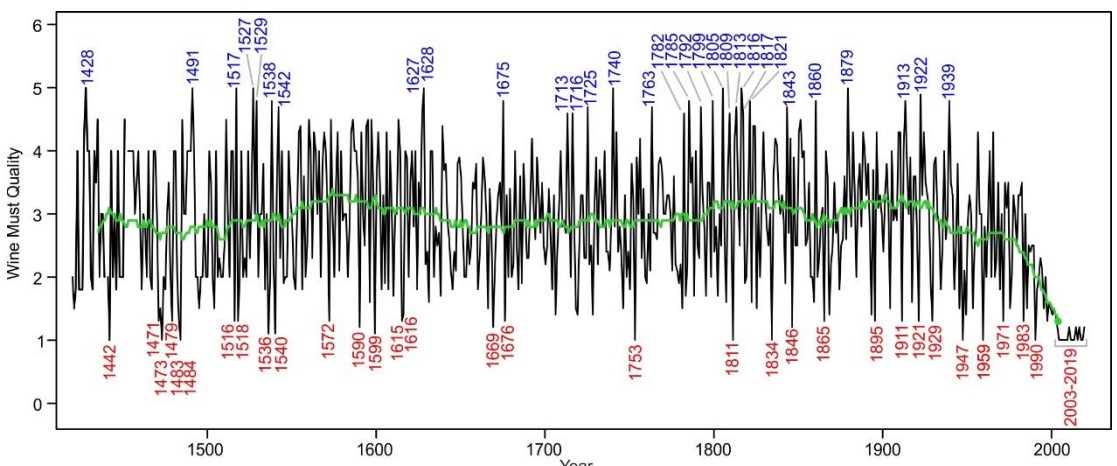

Figure 6. Annual average wine must quality in Western Europe 1420 to 2019
ranking from 1 top quality to 5 poor quality. The multidecadal trend is highlighted
by a 31-year low pass filter (green). The 5% top wine musts and the 5%
poorest ones are indicated in red and in blue, respectively. The almost complete
lack of top wine must qualities between 1676 and 1811 and the high frequency
of poor qualities between 1782 and 1821 is striking.





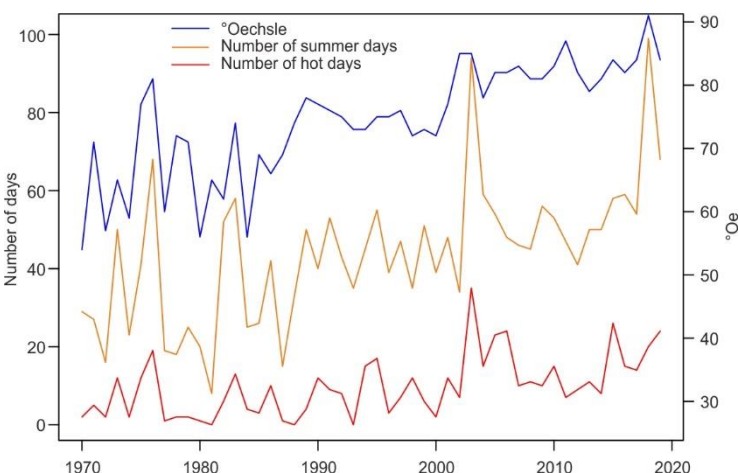

Figure 7. Riesling must quality, number of summer-
days and hot days in Remich (Lu) 1970–2019
Summer days: Days reaching 25°C. Hot days
(reaching 30° C).
Data: Oechsle density of the Riesling cultivar
and temperature measurements
at the viticultural research station of Remich (Lu)
Data: Mostwägungen, Luxemburg 1970–2019.