# Peer review of "years of wine must quality and April to August temperatures in Western Europe 1420–2019"

_Climate of the Past, 2023_

## Author Comment (AC1)

**The goal of the study is to determine if wine must quality can be used to obtain useful information on summer temperature in Europe over the past centuries. This is highly relevant as uncertainties remain on past climate variations during this period and new high-quality reconstructions are welcome.** The paper describes well the basis of the link between must quality and climate, the available long series and make a strong case on the interest of wine must quality as a proxy of past summer conditions. I thus recommend the publication of this study after some minor changes as detailed below.

General point

1/ In the majority of studies devoted to new records covering the past centuries, a reconstruction of a physical variable (like for instance here summer temperature) is proposed and validated against modern observations. This is not explicitly the case here. The dependence of the must quality on temperature and precipitation is discussed (e.g. Table 7) but this is not used to provide a reconstruction. think the authors should justify this choice and explain why they decide not to show such a reconstruction**.**

**This important issue might be discussed at the beginning of chapter 3 "Methods": We propose the following text: "A validation of the data against modern climate observations was not attempted because the data on wine must quality between 1881 and 1989 are of lower quality that those between 1751 and 1880 (Table 4). Starting in 1990 the quality values are no longer meaningful in terms of climate history because of a substantial quality improvement. In the field of climate reconstruction, data assimilation methods (or other inverse methods) are increasingly used, and these methods require a forward model such as the statistical model presented in this paper. The inversion approaches range from simple Bayesian methods such as weighted analogs sampled from climate model simulations (Reichen et al., 2022) to off-line Ensemble Kalman Filters (Valler et al., 2022). The series shown can be used in any of these approaches, together with other series, to obtain a climate reconstruction. We will give more explanations on this in the revised manuscript.**

The correlations of must quality with tree ring MXD do not show a clear pattern. They are generally lower prior to 1880 and at the same level between 1881 and 1989, without being able to provide any justification for this. The correlation of mean quality with GHD is considerably lower between 1881 and 1989 than in the two preceding time periods, while the correlation of GHD+ with tree ring MXD is at the same level.

**This points to inaccurate assessment of vine must quality due to low quality in face of the breakdown of the harvest amount which in Switzerland coincided with a period of transition from expert evaluation to density measurement. For Luxembourg where the transition took place starting in the mid nineteenth century, correlations are higher (not shown).**

**Specific points**

1/ Table 4 provides the correlation of must quality and GHD +, tree ring MXD and GHD+ but surprisingly to me not Tree Ring MXD and must quality. Is there a reason for this choice? As

must quality is the topic of this paper, this would have been instructive to see how the series compares to the one of tree ring MXD and if this agreement is higher or lower than between Tree Ring MXD and GHD+.

**The correlation of wine must quality and Tree Ring MXD was included in Table 4. We propose the following interpretation: "The correlation of must quality with tree ring MXD do not show a clear pattern. It is generally lower prior to 1880 and remains at the same level between 1881 and 1989. No interpretation is provided for this. The correlation of mean quality with GHD+ is lower between 1881 and 1989 than in the two preceding time periods, while the correlation of GHD+ with tree ring MXD is at the same level. This points to inaccurate assessment of vine must quality due to low quality in face of the breakdown of the harvest amount which in Switzerland coincided with a long period of transition from expert evaluation to density measurement. For Luxembourg where the transition took place starting in the mid nineteenth century, correlations are higher (not shown)."**

2/ Line 217 and 225. I have not understood to what refer the 'small number of cases per year' and the 'low case numbers'. If the series are annual, each year has just one number for me. Is this due to missing years? Is it related to the small number of sources for this period (table 1)?

**The term cases is misleading in this context. We will replace it by "observations".**

3/ Title of section 4: I would not use 'The climate model' because it is a statistical model of wine quality that is presented, not a climate model, even if it uses meteorological variables.-

**We will us "statistical model" throughout the paper.**

4/ Line 272. Is there a reason why precipitation in April and August are important and not the other months? Were the other months well below the selection criteria or were they close to be accepted by the backward selection?

**The other months were below the selection criteria. The two months are far apart such that the influences are likely different.**

5/ Line 287. When the 'same regression model' is mentioned for GHD, I guess it means the same meteorological variables but it has been recalibrated for GHD. The much higher explained variance for the verification period suggests that the behavior of this record is more stable in time than the one of must quality.

**Yes, a model with the same variables was calibrated against GHD. We will be more precise and will add additional discussion.**

6/ For figure 5, 5 years of high must quality and 5 years of low quality seems a small sample to me. Are the conclusions robust if a larger number of years is selected?

**The years were selected according to specific thresholds. A slight increase in the threshold would greatly increase the number of cases thus weakening the robustness of the conclusion.**

7/ Line 298. I guess 19 and 20 refers to the number of days. I would add the information explicitly.-

**the text is on p.199. We will modify the text accordingly**

8/ Figure 6 presents the same series as figure 4 if I am right but with a different caption. This could introduce confusion. I would include ll the information for figure 4 and then explain in figure 6 that the same time series is shown.-

**-accepted**

9/ Lines 321-322. Is there a source or reference for those nicknames?

**Müller 1953**

10/ Line 343. The information on the size of grape harvest seems a bit out of context here. It is required to have a longer discussion, including ideally references so that the reader could have an idea of the potential of this variable for climate reconstructions.-

**This issue will be tackled in a subsequent article-. A longer discussion would indeed be out of context.**

Another issue:
Do you agree that we approximately include your general evaluation in the abstract:

**"This is highly relevant as uncertainties remain on past climate variations during this period and new high-quality reconstructions are welcome."**

---

## Author Comment (AC2)

**Queries Reviewer 2**

The manuscript compares climate conditions to wine must (and wine) quality assessment series.

The paper is clearly written and quite straightforward.

The research present is of sound quality and seems to me very relevant in general. Addressing wine (wine must) quality / climate relationship is very welcome to me, as the subject is seldom consider long-time series.

**Here are some specific clarifications / questions I address to the authors**:

L40: Why using the first name of Andrea Kiss (et al.) here while ignoring them for the previously cited authors Pejml, Brazdil? Of course, you can choose the way you quote each authors, as long as they match the journal standard…it is only a genuine question here.  However, I would use "**their May-August temperature reconstruction**" rather than "her" because the paper cited has been written by 3 authors, and not only A. Kiss.

**Reply: We put instead: Kiss et al. (2011) included grape quality data from the late 18th century in their May–August temperature reconstructions for western Hungary.**

L51: I'm not sure www.winenium.com is the right citation to support epistemology. I suggest the citation of an English or latin dictionary.

**Reply: We refer instead to Wikipedia https://en.wikipedia.org/wiki/Must**

L79: Gregory V. Jones is a climatologist (https://en.wikipedia.org/wiki/Gregory_V._Jones)

**Reply: For clarification we prefer the original wording: "Gregory V. Jones is an American research climatologist specializing in the climatology of viticulture."**

L127: "This device, …" The device is not explained. Do you mean the hydrometer? Hasn't the first scale for density this device (also used for wine must) proposed by André Baumé ?

**Reply: We write instead "Appropriate devices"**

L221: downy mildew, phylloxera could be written without capital letters as the common name is used here.

**Reply : Accepted**

L235: Combe et al. (2015) is missing in the reference list. I would also stress that harvest (unless for sparkling wines) occurs much later than 3 / 4 weeks after veraison.  See for instance table 1 from https://doi.org/10.3354/cr01048

**Reply: It should read Combe and Smart (2015) which is in the list of references**.

L242: "The meteorological variables…..taken from the re-analysis (Valler et al., 2023)". Written as it is, it seems like "the re-analysis" is the name of a specific climate data series, while it is a generic name in climatology referring to historical climatological data gridded

field combining climatre data assimilation and modelling. I believe the sentence would be clearer providing directly the name of the product established in the cited submitted work : "ModE-RA re-analysis".

**Reply: We put "The meteorological variables taken from the rModE-RA re-analysis (Valler et al., 2023)"**

L298: This follows the expected pattern, except that one might also have suspected more abundant high-pressure days over central Europe (type 5) in the good quality years. We find almost the same number (19 versus 20). This is perhaps due to the fact that this type is rare.

**Reply: We put "We find almost the same number of #5 weather type in both vintages identified with either low or high must quality. This might be due to the fact that this type is rare."**

L313: What would be a standard price for Château d'Yquiem produced in the 1940s or 1950s?

**Reply: We don't know whether such evidence is available at all. We find that this it is not relevant to the topic of this article.**

L327: Why "wine musts obtained in the past 20 years must be classified top…" isn't using the verb "must" a little bit excessive here?

**Reply: Accepted. We put instead "most wine musts obtained in the past 20 years can be classified top"**

L330: I did not understand whether the authors consider Riesling or Muller-Thurgau grape varieties. These are two different cultivars (https://ojs.openagrar.de/index.php/VITIS/article/view/4608)

**Reply: Accepted. (Müller-Thurgau) is removed.**

L335: Do you mean grey mould (bunch rot)? While I don't have access to series of bunch rot in viticulture Switzerland, Germany and France from 1970 to 2019, it seems to me that bunch rot has not specifically developed in theses regions in the 21st century.

**Reply: Accepted. We just put "mould."**

---

## Author Response (AR1)

**List of revisions**

**General points:**

**Change of title: Most of the discussion deals with summer temperatures. There for we think that this title is more appropriate.**

1/ In the majority of studies devoted to new records covering the past centuries, a reconstruction of a physical variable (like for instance here summer temperature) is proposed and validated against modern observations. This is not explicitly the case here. The dependence of the must quality on temperature and precipitation is discussed (e.g. Table 7) but this is not used to provide a reconstruction.  think the authors should justify this choice and explain why they decide not to show such a reconstruction.

**Line 200 This important issue might be discussed at the beginning of chapter 4 "Methods": We propose the following text: "A validation of the data against modern climate observations was not attempted because the data on wine must quality between 1881 and 1989 are of lower quality that those between 1751 and 1880 (Table 4). Starting in 1990 the quality values are no longer meaningful in terms of climate history because of a substantial quality improvement. In the field of climate reconstruction, data assimilation methods (or other inverse methods) are increasingly used, and these methods require a forward model such as the statistical model presented in this paper. The inversion approaches range from simple Bayesian methods such as weighted analogs sampled from climate model simulations (Reichen et al., 2022) to off-line Ensemble Kalman Filters (Valler et al., 2022). The series shown can be used in any of these approaches, together with other series, to obtain a climate reconstruction.**

The correlations of must quality with tree ring MXD do not show a clear pattern. They are generally lower prior to 1880 and at the same level between 1881 and 1989, without being able to provide any justification for this. The correlation of mean quality with GHD is considerably lower between 1881 and 1989 than in the two preceding time periods, while the correlation of GHD+ with tree ring MXD is at the same level.

**Line 231 This points to inaccurate assessment of vine must quality due to low quality in face of the breakdown of the harvest amount which in Switzerland coincided with a period of transition from expert evaluation to density measurement. For Luxembourg where the transition took place starting in the mid nineteenth century, correlations are higher (not shown).**

**Specific points**

Line 40: We adopted to the proposed change putting "**Kiss et al. (2011) included grape quality data from the late 18th century in their May–August temperature reconstructions for western Hungary.**"

Line 51f.  Instead of "winnenium.com **we referred to Wikipedia** https://en.wikipedia.org/wiki/Must

Line 79f. We adopted to the proposed changes by putting for clarification the original wording: **"Gregory V. Jones is an American research climatologist specializing in the climatology of viticulture." (https://en.wikipedia.org/wiki/Gregory_V._Jones)**

Line 94 Additional input "Society for Wine History" (*Gesellschaft für Geschichte des Weines e.V.*)

Line 127: Instead of «this device" We put "Appropriate devices"

Line 130 Additional Input on source validity

Line 221 Accepted

2/ Line 217 and 225. I have not understood to what refer the 'small number of cases per year' and the 'low case numbers'. If the series are annual, each year has just one number for me. Is this due to missing years? Is it related to the small number of sources for this period (table 1)?

**The term cases is misleading in this context. We will replace it by "observations". O.K.**

Line 235 Missing reference "Combe et al. 2015", **we put instead "Combe and Smart (2015) " which is in the list of references.**

Line 242: Instead of "The meteorological variables», we put "**The meteorological variables taken from the rModE-RA re-analysis (Valler et al., 2023)**"

4/ Line 272. Is there a reason why precipitation in April and August are important and not the other months? **The other months were below the selection criteria. The two months are far apart such that the influences are likely different**.
We put: The presentation of the results begins with an overview of the climatic analysis and then discusses the final wine must quality series and its changes over time. The relation of wine must quality with climate was investigated with the regression model. The backward selection resulted in a statistical model that retained temperature and precipitation of all relevant months April to August. Temperatures in this period were all significant according to the results of Lauer, Frankenberg (1986) and Lorusso (2013), as well as precipitation in April and August. Precipitations in May, June and July were below the selection criteria. Admittedly, contemporary observers only discussed conditions in the summer months.

Line 251 The meteorological variables were taken from the ModE-RA re-analysis  (Valler et al., 2023)

5/ Line 287. Yes, a model with the same variables was calibrated against GHD. We will be more precise and will add additional discussion.

**We put: For comparison, we also applied the same regression model to grape harvest dates and found an even better performance (87% explained variance in the calibration, 73% in the verification period, see Table 7). Unlike wine must quality, GHD is rather insensitive to August temperature but shows even higher sensitivity to temperature in the preceding months, such that a higher explained variance results. Also sensitivity to precipitation is similar.**

7/ Line 298. I guess 19 and 20 refers to the number of days. I would add the information explicitly.-**Accepted**

Line 298: Unclear sentence. We put instead: "**We find almost the same number of #5 weather type in both vintages identified with either low or high must quality. This might be due to the fact, that this type is rare.**"

Line 313: "standard price for Château d'Yquiem produced in the 1940s or 1950s?"

**We don't know whether such evidence is available at all. After all we find that this issue is not relevant to the topic of this article**.

9/ Lines 321-322 reference for nicknames: **Müller 1953**

Line 327: Accepted. **We put instead "most wine musts obtained in the past 20 years can be classified top"**

Line 330: Riesling or Müller Thurgau? **Accepted. (Müller-Thurgau) is removed.**

Line 335: Accepted. **We just put "mould"**

10/ Line 343. The information on the size of grape harvest seems a bit out of context here. It is required to have a longer discussion, including ideally references so that the reader could have an idea of the potential of this variable for climate reconstructions

**This issue will be tackled in a subsequent article-. A longer discussion would be out of context.**

**Tables**

1/ Table 4 provides the correlation of must quality and GHD +, tree ring MXD and GHD+ but surprisingly to me not Tree Ring MXD and must quality. Is there a reason for this choice? As must quality is the topic of this paper, this would have been instructive to see how the series compares to the one of tree ring MXD and if this agreement is higher or lower than between Tree Ring MXD and GHD+.

**The correlation of wine must quality and Tree Ring MXD was included in Table 4. We propose the following interpretation: "The correlation of must quality with tree ring MXD do not show a clear pattern. It is generally lower prior to 1880 and remains at the same level between 1881 and 1989. No interpretation is provided for this. The correlation of mean quality with GHD+ is lower between 1881 and 1989 than in the two preceding time periods, while the correlation of GHD+ with tree ring MXD is at the same level. This points to inaccurate assessment of vine must quality due to low quality in face of the breakdown of the harvest amount which in Switzerland coincided with a long period of transition from expert evaluation to density measurement. For Luxembourg where the transition took place starting in the mid nineteenth century, correlations are higher (not shown)." O.K.**

8/ Figure 6 presents the same series as figure 4 if I am right but with a different caption. This could introduce confusion. I would include ll the information for figure 4 and then explain in figure 6 that the same time series is shown.- **-accepted**

**The data are identical to those în Figure 4, but are presented in a different context.**

**References:**

---

## Author Response (AR2)

Geographisches Institut, Hallerstr. 12, CH-3012 Bern

[Figure]

b
**UNIVERSITÄT
BERN**

Philosophisch-naturwissenschaftliche
Fakultät

**Geographisches Institut**

Bern, 3 May 2024

**Submission of "600 years of wine must quality and April to August temperatures in Western Europe 1420–2019"**

Dear Editor

Please find attached the paper: "600 years of wine must quality and April to August temperatures in Western Europe 1420–2019" by Christian Pfister, Stefan Brönnimann, Andreas Altwegg, Rudolf Brázdil, Laurent Litzenburger, Daniele Lorusso, and Thomas Pliemon published in *Climate of the Past*. We have incorporated the minor suggestions and are happy to submit the revised manuscript. The changes can be seen in the attached manuscript with track changes.

Thank you very much for handling our manuscript.

Best regards,

Stefan Brönnimann

Prof. Stefan Brönnimann          Tel. +41 031 631 88 85
Geographisches Institut          stefan.broennimann@giub.unibe.ch
Hallerstr. 12
CH-3012 Bern

[revised manuscript text omitted]

and the high frequency

695

[Figure]

**Figure 7.** Riesling must quality, number of summer-
reaching 25°C, h. Hot days: days

700    Summer days: Days reaching 25°C, h. Hot days: days
(reaching 30° C).
Data: Oechsle density of the Riesling cultivar
and temperature measurements
at the viticultural research station of Remich (Lu),
705    Data: Mostwägungen, Luxemburg 1970–2019.